# Viroids as Companions of a Professional Career

**DOI:** 10.3390/v11030245

**Published:** 2019-03-12

**Authors:** Núria Duran-Vila

**Affiliations:** Centro de Protección Vegetal y Biotecnología. Instituto Valenciano de Investigaciones Agrarias (IVIA), Moncada, 46113 Valencia, Spain; duranvila.nuria@gmail.com; Tel.: +34-690171150

**Keywords:** Viroid diseases, Symptom expression, Cooperation approaches

## Abstract

Since the early 1970s when “virus-like” agents were considered as the cause of two diseases (potato spindle tuber and citrus exocortis), their study and further characterization have been linked to the development and use of molecular biology tools. Sucrose density gradient centrifugation and polyacrylamide gel electrophoresis (PAGE) played a critical role in the pioneering studies of PSTVd and citrus exocortis viroid (CEVd). This was later modified by using other PAGEs (sequential PAGE, return PAGE, two-dimensional PAGE), and/or different staining methods (ethidium bromide, silver nitrate, etc.). Since then, disease-causing agents suspected to be viroids were usually subjected to a number of tests to define their: (i) Molecular nature (RNA or DNA; single stranded or double stranded; circular or linear RNA); (ii) molecular weight; (iii) secondary and tertiary structure. Further biological assays are also essential to establish the relationship of a viroid with plant disease and to fulfill Koch’s postulates.

## 1. Introduction

The discovery of viroids is associated with the available phytopathology approaches used to understand the cause of two diseases, potato spindle tuber and citrus exocortis. After its first description, the potato spindle tuber rapidly spread to several US states [1,2] known as “degeneration disease”. Citrus exocortis, also reported as “scalybutt”, was described as a bark scaling disorder affecting the rootstock of trees grafted on the trifoliate orange (*Poncirus trifoliate* (L.) Raf.) rootstock that resulted in dwarfed trees [3,4,5]. 

As a student in the 1970s, I was part of a team involved in the identification of the causal agent of the exocortis disease, a situation that provided the opportunity to observe first-hand how the development of molecular biology technologies were successfully used to learn more and more about the so-called “virus-like agents”, or VIROIDS. After transmission of the severe forms of the disease agent to the herbaceous host *Gynura aurantiaca* [6], the causal agent was isolated and characterized as a 371 nucleotide RNA molecule, the citrus exocortis viroid (CEV/CEVd) [7,8,9]. The development of the “Etrog” citron (*C. medica* L.) bioassay [10], with its greater sensitivity and rapid symptom development [11], led to the identification of additional symptoms, and field sources were frequently classified as mild, moderate, or severe, solely on the basis of the citron reaction.

The use of polyacrylamide gel electrophoresis (PAGE) was critical for the accomplishments of early studies. In addition, the implementation of slab gel electrophoresis instead of the tube electrophoresis and the new ethidium bromide staining allowed the simultaneous analysis of 10–15 samples instead of single samples with the tube gel electrophoresis and bromophenol blue staining commonly used at that time (Figure 1). However, with this technique, we were not able to see those that were believed to be mild and moderate strains of the same agent (viroid). Additional tools such as double PAGE (return PAGE, two-dimensional PAGE, sequential PAGE–sPAGE) were necessary to go a step further. It was the analysis of citrus hosts instead of herbaceous hosts by sPAGE and silver staining that provided the necessary information for the identification of viroids other than CEVd in what were thought to be mild and moderate strains of the exocortis agent. Several bands, possibly corresponding to viroid-like RNAs different from CEVd, were observed when most citron samples were assayed by sPAGE and silver staining (Figure 2). It took a great effort to produce citron plants containing only one of each of these potential viroid RNAs. These plants were essential to confirm that these RNAs were infectious (Figure 3) and to define the type of symptoms they induced in the Etrog citron indicator. The RNAs detected by these sPAGE assays were tentatively named RNA-I, RNA-II, and RNA-III [12]. Further characterization based on the available technologies (electrophoretic mobility, CF11 cellulose chromatography, and hybridization to cDNA probes) and the association of each RNA showing different electrophoretic mobility with respect to CEVd with specific symptoms on Etrog citron allowed for the organization of these RNAs into four groups predicted to correspond to four different viroids that were termed CVd-I to IV (CVd-I, CVd-II, CVd-III, CVd-IV) [13]. More recently, three additional viroids have been identified in citrus crops and tentatively termed CVd-V [14,15], CVd-VI (former CVd-OS) [16,17], and CVd-VII [18] (Table 1).

With the availability of single viroid samples, it was observed that CVd-II was not homogeneous but conformed by several bands with slightly distinct electrophoretic mobilities (Figure 3) that were further characterized as specific variants of *Hop stunt viroid* (HSVd) [19]. In terms of their biological activity, only those with faster mobilities, initially termed as *Citrus cachexia viroid* (CCaVd) [20] were able to cause an important disease named CACHEXIA [21] that had been reported earlier as XYLOPOROSIS [22]. Further characterization was accomplished when RT-PCR, cloning, sequencing, and easy to handle technologies became available, with which we were able to show that pathogenicity was associated with a specific conformation of the V domain due to a set of 5–6 specific changes of the rod-like secondary structure [23,24].

A similar situation regarding the occurrence of several viroids infecting a crop was found in grapevines even when no symptoms were observed [25,26,27]. This unique prevalence of multiple viroid infections in crops of economic importance such as grapevines and citrus, both originated in South East Asia and vegetatively propagated for several centuries, poses the question that viroids may not be necessarily deleterious or even may have provided and still provide desirable properties. If that was the case, infected plants may have been unconsciously selected for further propagation.

Even though viroids have usually been identified because of the symptoms they induce in their host plants, they are likely to have prevailed and evolved since their origin in symptomless wild species. However, agricultural practices have promoted their prevalence, especially in vegetatively propagated species, a situation that promoted the development and application of specific control measures. 

## 2. In Vitro Tissue Culture Approaches

As already indicated, early descriptions and approaches were always associated with the availability of easy to handle herbaceous plants showing characteristic symptoms of stunting, epinasty, and leaf rugosity. Early viroid studies were also concomitant with the development of in vitro tissue culture techniques that attracted the attention of plant scientists because they were easy-to-handle and available under laboratory conditions. In vitro tissue cultures involved the growth of isolated roots, callus, and cell suspension cultures, and the use of isolated protoplasts was even more relevant, especially when it was shown that they could easily regenerate a cell wall and divide [28,29]. With the demonstration that macromolecules and viruses could be artificially introduced into intact protoplasts, these were envisaged as a desirable tool for plant virology studies. Unfortunately, the fact that different viruses held different properties in terms of their presence and concentration in tissues, callus, cells, and protoplasts resulted in unexpected and frustrating results. However, different approaches regarding the effect of CEVd infection provided interesting insights that are probably related to the biological changes associated with symptom expression.

The inability of CEVd containing tissues to respond to exogenous auxins [30] as well as the high rates of ethylene released by CEVd infected cells [31] were considered to be probably associated to the molecular events responsible for symptom expression. In vitro tissue culture approaches also revealed differences in cell wall composition and structure that were probably limiting tissue expansion, and in the case of CEVd, infected tissues remain more compacted [32,33]. However, in the case of cell cultures which did not show morphological changes due to viroid infection, they were found to have efficient growth patterns in terms of longevity, biomass accumulation, and enhanced tolerance to certain sugars [34], all of the positive traits associated with viroid infection.

The most relevant input of in vitro tissue culture technologies was the development of shoot-tip culture and shoot-tip grafting technologies as widely used approaches to recover viroid-free plants from viroid-infected sources while maintaining biological/genetic properties of the source plant/cultivar. These approaches are based on the accessibility of some viroids in the apical meristem and are impaired due to the absence of vascular tissues and/or the result of RNA-silencing phenomena. Therefore, they are being used to recover viroid-free plants from herbaceous as well as woody species [35,36,37], as well as to separate viroid species and viroid populations from multiple infections [38].

## 3. Field Assays Supporting Viroid Characterization

Soon after the identification of several viroids being widespread in commercial citrus, we became aware that Koch’s postulates had not even been fulfilled in the case of the viroid defined as the causal agent of exocortis (CEVd). The first description of the exocortis disease by Fawcet and Klotz in 1948 [3] referred to a bark scaling disorder in the trifoliate orange (*Poncirus trifoliata* (L.) Raf.) rootstock but it had not been demonstrated if CEVd actually caused this type of symptoms. In addition, from the agronomical point of view, it was relevant to find out which was the effect of viroid infection on the field performance of citrus trees. However, to elucidate all this was not an easy task because it required long-term field assays which were expensive and not easy to handle. “Long-term field assays” meant to keep inoculated plants/trees in the field for 10–15 years in order to get information on symptom development and their effect on crop production and fruit quality. The establishment of field assays was not straightforward because trifoliate orange requires acidic soils that are extremely rare in the citrus growing areas of Spain. We were lucky that Prof. J.M. Bové put us in contact with Prof. R. Vogel at the Saint Guliano Station in Corsica (where clementine is the most important citrus crop and trifoliate orange the most common rootstock) and we initiated a successful collaboration with his team. Graft inoculation in the greenhouse was done in May 1989 and included several isolates of the five citrus viroids (CEVd, CBLVd, HSVd, CV-III, and CVd-IV) known at that time, with treatments that included single viroid infections as well as all of their possible combinations. Inoculation of trifoliate orange seedlings was followed by graft propagation of clementine buds and the inoculated/infected trees were transplanted in the field the following year. After yearly data collection, the assay was terminated in 2002 [39,40]. The results confirmed that CEVd and specific sequence variants of HSVd are the causal agents of exocortis and cachexia, respectively. Other characteristic symptoms such as bark cracking, bumps, and green streaks in the wood, were also identified (Figure 4). Additional information regarding dwarfing and yield reduction, as well as changes in the expected performance as a consequence of synergistic and antagonistic viroid interactions, were also identified. The information generated in that assay supported the present nomenclature of CVd-III as *Citrus dwarfing viroid* (CDVd) and CVd-IV as *Citrus bark cracking viroid* (CBCVd) (Table 1).

Since in Spain and many other citrus growing areas, the major rootstock is not trifoliate orange but is Citrange hybrids (*Citrus sinensis* × *P. trifoliata*), additional field assays were also established to define the performance of viroid infected trees grafted on such rootstocks. Symptom evaluation was complex because trees grafted on these rootstocks were more tolerant of viroid infection, and the environmental conditions (mainly temperature) did not enhance symptom expression (Figure 4). By pulling out the trees, we were able to see/evaluate mild symptoms as well as the effect of viroid infection on the root size and therefore canopy size [41,42]. Such results were also confirmed by evaluating the performance of viroid infected trees in commercial plantations [43].

Further field information was achieved through an assay conducted in Tunisia to evaluate the effect of eight rootstocks on the behavior of the most important cultivar grown in Tunisia, the sweet orange “Maltaise demi sanguine” [44]. The rootstocks chosen were Cleopatra mandarin (*C. reshni* Hort. Ex Tan.), trifoliate orange, Swingle citrumelo (*C. paradisi* × *P. trifoliata)*, Rangpur lime (*C. limonia* Osb.), volkamer lemon (*C. volkameriana* Ten. And Pasq.), sour orange (*C. aurantium* L.), alemow (*C. macrophylla* Webster), and Carrizo citrange, and as a result of the identification and widespread occurrence of viroids in commercial plantations [45], their effect was also taken into consideration.

## 4. Additional Hosts, New Viroids, and Viroid Variants

Searching for viroids in vegetable species revealed that CEVd was present and highly widespread in many herbaceous crops. In some instances such species were symptomless, a situation that enhanced the survival and spread of CEVd. In other instances, such as the case of a CEVd variant found in India, as naturally infecting commercial tomato plantings, infected plants displayed apical proliferation, stunting, epinasty, leaf distortion, and vein necrosis [46]. However, natural CEVd infections have also been found in symptomless grapevine, tomato, eggplant, turnip, carrot, and broad bean [27,47,48,49]. This widespread occurrence fits with the quasi-species model characteristic of the heterogeneous populations of viroid variants helping host selection of variants that are best adapted to a given host and their survival when transmitted to other hosts. Eggplant was revealed as an unusual symptomless CEVd host that allowed the generation of stable 467-nt variant with a 96-nt duplication of the right terminal region (CEVd-D96) similar to the CEVd-D92 variant characterized previously in a hybrid tomato [50,51]. These findings illustrate the generation of enlarged sequences even outside the family *Avsunviroidae*.

The search for viroids in vegetable crop species revealed the presence of a new viroid in symptomless eggplant. The viroid was characterized as being horizontally seed transmitted as an RNA with a minimal free energy structure that allowed the formation of stable hammerhead structures/ribozymes characteristic of members of the *Avsunviroidae* family. With the available information, this viroid was named *Eggplant latent viroid,* and it is the type species of the genus *Elaviroid* within the family *Avsunviroidae* [52].

## 5. Personality Traits of Scientists: Competition vs. Cooperation

As in the case of other disciplines, viroid research requires the input of competent scientists. However, within the research world, there is a mixture of two quite different personality traits—competence and competition. The acknowledgment and recognition of the discovery of viroids put together two outstanding personalities, T.O.Diener and J.S.Semancik, who were attempting to identify the causal agent of two diseases, “potato spindle tuber” and “citrus exocortis”. These two competent scientists were extremely competitive, and they did not communicate with each other, a situation that has created misunderstandings and questionable actions. Today, in many publications, T.O.Diener is considered the first one to have discovered/described VIROIDS as a new class of pathogenic agents affecting potatoes in a publication dated in 1971 [53], whereas similar work of J.S.Semancik working on citrus published in 1972 [7], in many instances remains unknown. As illustrated (Figure 5), these findings were the result of independent approaches and the two articles were accepted for publication within a six-month period. Therefore, these two scientists should share the same recognition. 

Having closely seen the generation and evolution of new knowledge, I developed my carrier under the principles of collaboration and cooperation. The application of these principles became the link between viroid research and many issues of my personal life in different parts of the world (Figure 6).

## Figures and Tables

**Figure 1 viruses-11-00245-f001:**
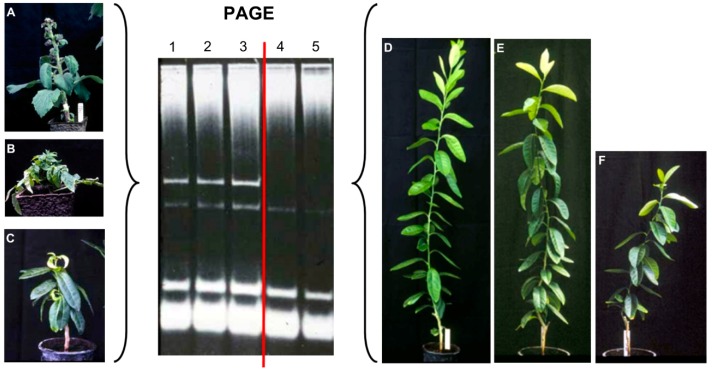
Analysis of different inoculated hosts by polyacrilamide gel electrophoresis (PAGE) and ethidium bromide staining. A characteristic band corresponding to a viroid RNA was stained in samples (1–3) from *Gynura aurantiaca* (**A**), tomato (**B**), and Etrog citron showing severe symptoms, whereas the same band was not observed in samples (4,5) from Etrog citron plants displaying mild and moderate symptoms (**D**–**F**).

**Figure 2 viruses-11-00245-f002:**
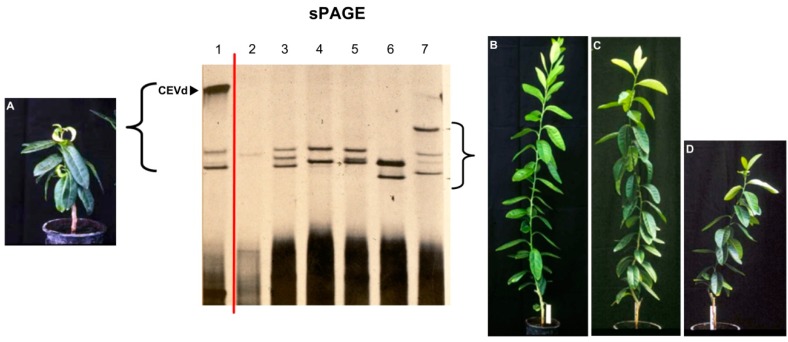
Analysis of inoculated Etrog citron plants by sPAGE and silver staining. Only plants showing severe symptoms (**A**) contained a band with the characteristic mobility of citrus exocortis viroid (CEVd) (1). Analysis of Etrog citron plants displaying mild and moderate symptoms (**B**–**D**) revealed bands with electrophoretic mobility different from CEVd (2–7). Virtually all plants contained a mixture of viroid-like RNAs that had to be further characterized.

**Figure 3 viruses-11-00245-f003:**
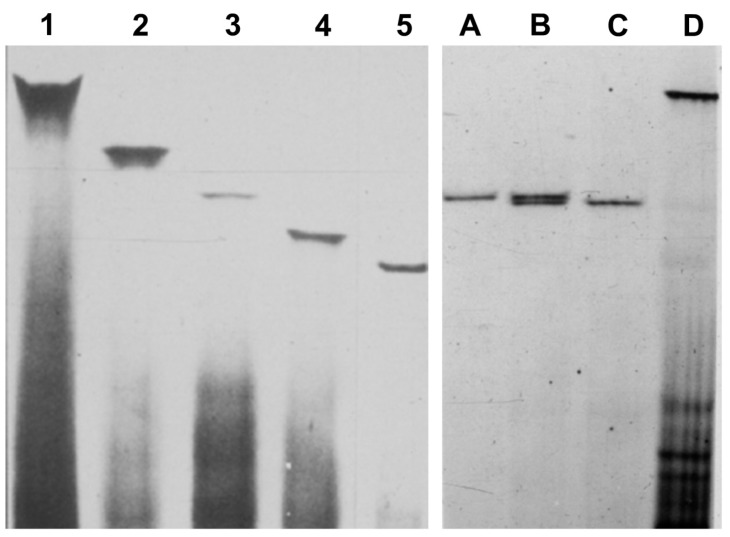
Analysis of inoculated Etrog citron plants by sPAGE and silver staining showing that they were infected with single viroids (1–5). CVd-II was not homogeneous but conformed by several bands (**A**,**C**) with slightly distinct electrophoretic mobilities (**B**) that were further characterized as specific variants of *Hop stunt viroid* (HSVd) showing faster mobility than CEVd (**D**).

**Figure 4 viruses-11-00245-f004:**
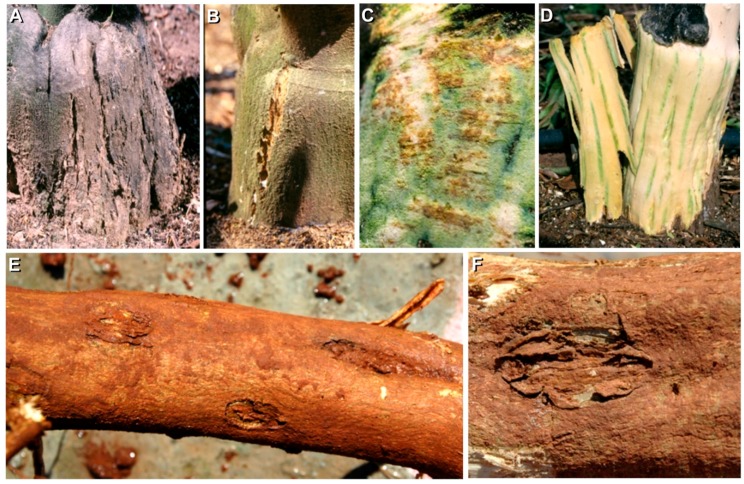
Characteristic symptoms as a result of viroid infection in plants grafted on trifoliate orange: Bark scaling induced by CEVd in trifoliate orange (**A**); bark cracking induced by *Citrus bark cracking viroid* (CBCVd) in trifoliate orange (**B**); wood pitting and gum deposits in the clementine scion caused by cachexia strains of HSVd (**C**); green streaks under the cracks induced by CBCVd in trifoliate orange (**D**). Lesions in the roots caused by CEVd in Citrange carrizo roots (**E**,**F**).

**Figure 5 viruses-11-00245-f005:**
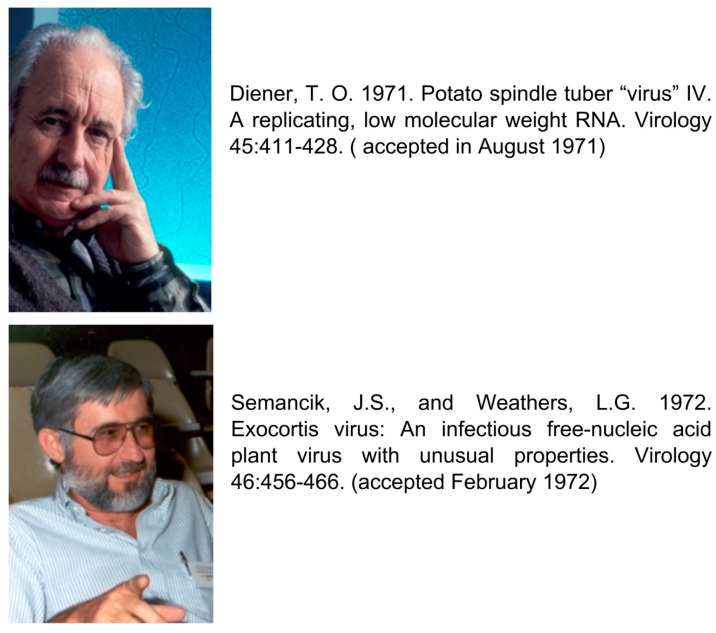
The two scientists responsible for the early discovery/characterization of viroids as disease-causing agents and their conflictive publication.

**Figure 6 viruses-11-00245-f006:**
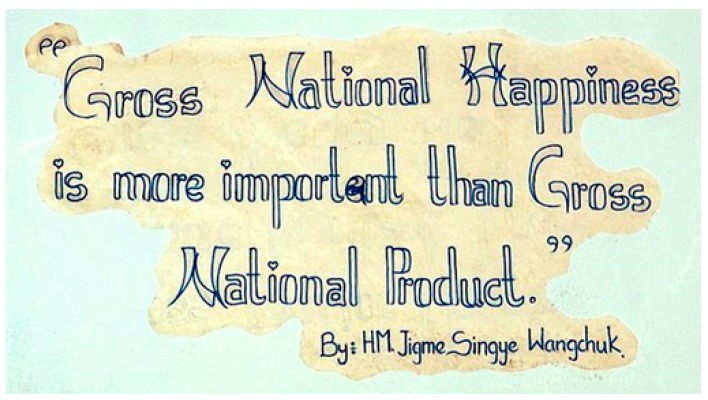
The dedication to cooperation issues in different parts of the world showed that in certain countries, as in Bhutan, the philosophy of life is to be taken into consideration.

**Table 1 viruses-11-00245-t001:** Citrus viroids and viroid species.

Family	Genus	Former Name	Species
*Pospiviroidae*	Pospiviroid	CEVd	*Citrus exocortis viroid* (CEVd)
	Hostuviroid	CVd-II	*Hop stunt viroid* (HSVd)
	Cocadviroid	CVd-IV	*Citrus bark cracking viroid* (CBCVd)
	Apscaviroid	CVd-I	*Citrus bent leaf viroid* (CBLVd)
	Apscaviroid	CVd-III	*Citrus dwarfing viroid* (CDVd)
	Apscaviroid		CVd-V
	Apscaviroid		CVd-VI
	Apscaviroid		CVd-VII

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
