# Peer review of "Viroids as Companions of a Professional Career"

_viruses, 2019, doi:10.3390/v11030245_

Reviewer 1 Report

I had some difficulty to gasp whether we had to review a commentary or a review article. I believe this is to some extent a weakness of this manuscript since personal comments succeed pure scientific descriptions in (what to this reviewer seemed) a rather awkward manner. Nevertheless, this is a very valuable article that provides the reader a rare, and precious encounter of the early events in viroid research.

Major comments:

the last paragraph, “5. Personality traits…” comes totally out of the blue and with hardly any connection to the previous. Although this reviewer is in agreement with all stated there, there needs to be some connection with the rest of the narrative.

Minor comments:

Line 100, "...easy-to-handle..." would reads better

Line 151,  “…(where Clementine IS the the most important…”

Line 190  “…highly widespread IN many herbaceous crops”

Line 22  “first TO have discovered/described VIROIDS as a new class…”

Author Response

The manuscript was submitted as a commentary and not a review. This is the reason to have included personal comments which is a rather unusual possibility. 

I appreciate the minor comments regarding mistakes that have already been corrected in the revised version.

Reviewer 2 Report

This is a very interesting review on the relevance of biological, molecular and in vitro approaches in the study of viroids. It is a personal focus that highlights the efforts and the r contribution of a dedicated and competent researcher to the knowledge we have today of viroids. I strongly suggest to accept this manuscript for publication. However, I also suggest to take into consideration the comments below that aims to improve the clarity of some sentences.

My personal congratulations to the author.

L15 and L16: I suggest to avoid the use of ellipsis. Please,  add other PAGEs or staining methods, if needed and remove the ellipsis

L18: maybe better to replace “to define: i).......” with “to define their i) molecular nature (RNA or DNA; single-stranded RNA or double-stranded RNA; circular or linear RNA), ii) molecular weight, ii) secondary and tertiary structure.

L57-69. Possibly, at L57, the author wants to refer to “this analysis” (sPAGE and silver staining) instead of “the first analyses”. If this is the case correct accordingly.

Moreover, several sentences should be revised in this section. In fact, I have concerns regarding the use of the term “band” (i.e.: most samples contained several bands, citron plants containing a single band, the observed bands were infectious). I do not believe that a sample contains a band and a band can be infectious. The author should be more precise in this respect. A suggestion for improve the clarity of this paragraph:

Several bands, possibly corresponding to viroid-like RNAs different from CEVd, were observed when most citron samples were assayed by sPAGE and silver staining (Fig. 2). It took a great effort to produce citron plants containing only one of each of these potential viroid RNAs. These plants were essential to confirm that these RNAs were infectious (Fig. 3) and to define the type of symptoms they induce in the Etrog citron indicator. The RNAs detected by these sPAGE assays were tentatively named RNA-I, RNA-II and RNA-III [12]. Further characterization based on the available technologies (electrophoretic mobility, CF11 cellulose chromatography and hybridization to cDNA probes) and the association of each RNA showing different electrophoretic mobility with respect to CEVd with specific symptoms on Etrog citron allowed the organization of the RNAs with different mobility with respect to CEVd into four groups predicted to correspond to four different viroids that were termed citrus viroid I to IV (CVd-I, CVd-II, CVd-III, CVd-IV) [13].

L80-81: Why the name of the diseases are capitalized? Generally, they are not.

L86-87: The sentence is unclear: is the author referring to HSVd variants of different length found in grapevine or to viroids of different length found in a single host plant? Please clarify.

L87-92: I think that the author is right, but in the context considered here, the absence of symptoms in the infected plants should be highlighted.

L99: Replace “As already indicated, early descriptions and approaches” with “ As already indicated, early descriptions of viroids and experimental approaches to study them “.

L129: I suggest to replace “viroids are no present in the apical meristem” with “accessibility of some viroids in the apical meristem is impaired due to”. In fact the presence of viroids in apical meristem has been shown in some viroid/host combination and during some specific developmental stage of the host.

L151: replace “in” with “is”

L190: replace “and” with “in”

L203: replace Avsunviroid family with “family Avsunviroidae” (in italics)

L205: replace “symptomless plants” with  symptomless eggplant

L222: some problems here: “VIRan approaach OIDS” should be “VIROIDS”

L224: replace “with a publication dated in 1972” with “ published in 1972”

L387-390 (Legend to Figure 1): I suggest to correct the second sentence as follows: “A characteristic band corresponding to a viroid RNA was stained in samples (1, 2, 3) from Gynura aurantiaca (A), tomato (B) and Etrog citron showing severe symptoms, whereas the same band was not observed in samples (4, 5) from Etrog citron plants displaying mild and moderate symptoms (D, E, F).”

L391-396 (Legend to Figure 2): The legend should better clarify the meaning of the panels B, C and D. Possibly, sPAGE of samples from these plants revealed bands with electrophoretic mobility different from CEVd. This notion should be clarified and the relationship between panels A to D and the samples analysed by sPAGE should also be indicated (as it was done in the legend to Figure 1).

Line 71: Table 1: The name of species must not be abbreviated. It must be written in extenso and with the first letter capitalized: i.e. Citrus exocortis viroid. The name of viruses must be written with the first letter non-capitalized and in normal style: i.e. citrus exocortis viroid, citrus viroid II, etc.

L399-404 (Legend to Figure 3). What is the sample in the lane D?

L407-408 (Legend to Figure 4). Which viroid was associated with the symptoms reported in each Panel?

Author Response

I really appreciate the input of this reviewer in terms of identification errors and suggesting changes to improve and clarify the manuscript.

All the suggestionshave been incorporated.